# Two-dimensional electrons at mirror and twistronic twin boundaries in van der Waals ferroelectrics

James G. McHugh [1,2], Xue Li[1,2], Isaac Soltero [1,2] & Vladimir I. Fal'ko [1,2] ✉

Semiconducting transition metal dichalcogenides ($MX_2$) occur in 2H and rhombohedral (3R) polytypes, respectively distinguished by anti-parallel and parallel orientation of consecutive monolayer lattices. In its bulk form, $3R\text{-}MX_2$ is ferroelectric, hosting an out-of-plane electric polarisation, the direction of which is dictated by stacking. Here, we predict that twin boundaries, separating adjacent polarisation domains with reversed built-in electric fields, are able to host two-dimensional electrons and holes with an areal density reaching ~ $10^{13} cm^{-2}$. Our modelling suggests that n-doped twin boundaries have a more promising binding energy than p-doped ones, whereas hole accumulation is stable at external surfaces of a twinned film. We also propose that assembling pairs of mono-twin films with a 'magic' twist angle $\theta^*$ that provides commensurability between the moiré pattern at the interface and the accumulated carrier density, should promote a regime of strongly correlated states of electrons, such as Wigner crystals, and we specify the values of $\theta^*$ for homo- and heterostructures of various TMDs.

Interfacial ferroelectricity[1] has recently been identified as a peculiar feature of the rhombohedral polytype of semiconducting transition metal dichalcogenides (TMDs: $MoS_2$, $WS_2$, $MoSe_2$, $WSe_2$, $MoTe_2$)[2-4]. For example, $3R\text{-}MX_2$ are layered van der Waals (vdW) crystals that have neither inversion nor mirror symmetry[5], permitting a c-axis ferroelectric (FE) polarisation, which is, indeed, generated by the interlayer charge transfer induced by weak hybridisation of chalcogen orbitals[6-10]. The direction of such polarisation is determined by the stacking of consecutive layers, identified as metal-over-chalcogen (MX) and chalcogen-over-metal (XM) configurations (see Fig. 1). The above relation between stacking order and FE polarisation, is demonstrated in a number of experiments involving TMD bilayers where polarisation inversion was produced by sliding of adjacent TMD monolayers[1,11,12], also suggests better stability of FE domains in vdW ferroelectrics, as compared to conventional FE crystals[13]. Potential steps of $+\Delta$ or $-\Delta$, for MX or XM domains, respectively (caused by the double layer of charge at the interface of consecutive monolayers), generate staircase potential profiles inside these domains[14], Fig. 1, which can be associated with a built-in electric field $\pm E_{FE}$ inside MX and XM twins.

In general, a piece of bulk FE material would contain domains of alternating electrical polarisation, separated by domain walls[13]. In ferroelectric $MX_2$, polarisation domains are nothing but structural MX- and XM-stacking twins, separated by mirror twin boundaries (mTBs). For electrons and holes, these mTBs represent maxima and minima of quantum wells hosting two-dimensional (2D) electrons or holes (which can, for example, be separated upon photo-excitation). Here, we predict that such twin boundaries can hold charge carriers with a density of up to $n \approx (0.7-0.9) \times 10^{13} cm^{-2}$, which can be achieved by photo-doping at cryogenic temperatures[15,16], and perform self-consistent analysis of the binding energies and quantum well wave functions to identify the twin structures which are most favourable for the formation of 2D electron layers in thin films of rhombohedral $3R\text{-}MX_2$ (M = Mo, W and X = S, Se, Te).

## Results and discussion
To describe the FE potential profile in a twinned bulk $3R\text{-}MX_2$, we have performed ab initio DFT, implemented in the quantum ESPRESSO package[17,18], analysing bulk supercells consisting of up to 12 layers per

[1]Department of Physics and Astronomy, University of Manchester, Oxford Road, Manchester M13 9PL, UK. [2]National Graphene Institute, University of Manchester, Booth St. E., Manchester M13 9PL, UK. ✉e-mail: vladimir.falko@manchester.ac.uk

twin (for sample relaxed structures, see Supplementary Data 1). Plane-wave kinetic energy cutoffs of 80 and 800 Ry were applied for wave-function and charge density expansions, respectively, and the Brillouin zone was sampled using a Monkhorst–Pack uniform $22 \times 22 \times 1$ $k$-point grid for all structures. The exchange-correlation effects were approximated by the generalised gradient approximation (GGA) using the Perdew–Burke–Ernzerhof (PBE) functional[19]. Ultrasoft pseudopotentials were used to approximate the interaction between the nucleus and electrons[20,21], and the vdW-DF2-C09 functional was implemented to compute interlayer adhesion for relaxing the lattice[22–25], using the Broyden–Fletcher–Goldfarb–Shanno (BFGS) quasi-Newton algorithm. Non-collinear spin–orbit coupling (SOC) was incorporated into all band structure calculations via ultra-soft fully relativistic pseudopotentials[26]. In particular, this gives us the layer-resolved, local electrostatic (ionic and Hartree) potentials across the studied structures (examples shown in Supplementary Fig. 1), consistent with the individual 'double-charge-layer' potential steps found in bilayers[2,10] and few-layer mixed-stacking crystals[14].

From this, we evaluate the intrinsic ferroelectric field in 3R-MoX₂, $\mathbf{E}_{FE}$ (as the envelope of the on-layer potentials) and estimate the 2D charge that would compensate for the intrinsic field $\mathbf{E}_{FE}$ (see Table 1). This sets up an electrostatic limit for the maximum carrier density that can be held by the twin boundary,

$$n = 2|\mathbf{E}_{FE}|\varepsilon_{zz}\varepsilon_0/e. \tag{1}$$

Here, $\varepsilon_{zz}$ is the $c$-axis (out-of-plane) dielectric constant of TMD (6.1 in MoS₂, 5.8 in WS₂, 7.3 in MoSe₂, 7.2 in WSe₂, and 10.7 in MoTe₂, according to refs. [14,27], $\varepsilon_0$ is the vacuum electric permittivity, and $e$ is the electron charge, leading to the density values listed in Table 1. We note that mono-twin film surfaces would hold maximum carrier density $\pm \frac{n}{2}$, so

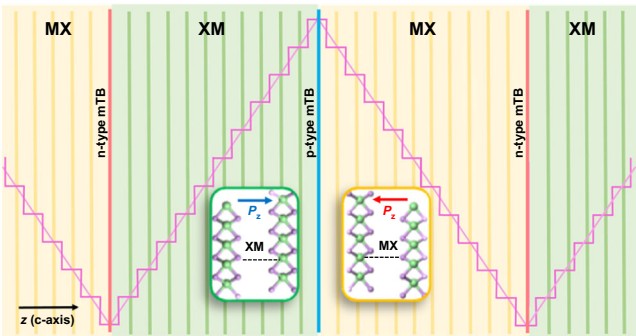

**Fig. 1 | Mirror twin boundaries in ferroelectric TMDs.** Schematic of mirror twin boundaries separating adjacent ferroelectric domains in 3R-TMDs (with MX- and XM-stacking identified in the insets), coloured green/yellow according to stacking. The staircase of potential energy steps for electrons ±Δ, and its linear interpolation is indicated by the purple profile. Twin boundaries are distinguished by their ability to accumulate electrons (n-type), or holes (p-type).

that internal mTBs follow a rule of sum for surface layer densities of the constituent twinned crystals, as sketched in Fig. 2a, b (and Supplementary Fig. 2). In addition, one can mechanically assemble two FE films, thus, creating a twisted twin boundary (tTB): such an interface would hold the same maximum charge carrier density as an untwisted mTB. To mention, a thick MX (or XM) monotwin film with $N_{layers} \gg E_g/\Delta \sim 20$–30 for the TMDs studied here (where $E_g$ is the semiconductor bandgap) would self-dope its surfaces to the same densities as illustrated in Fig. 2a. Similarly, a multidomain crystal with such thick twins would self-dope with a density distribution as in Fig. 2b.

To determine self-consistent binding energies and, then, to compare those with the Fermi level of carriers with the accumulated density, we computed the effective masses ($m_x$, $m_y$, $m_z$) for Q-point electrons in the conduction band and $\Gamma$-point valence band holes (and spin–orbit splitting, $\Delta_{SO}$, for the Q-point band edge). The in-plane masses were found by fitting the band edge energy profile (e.g., in Supplementary Fig. 3) to a parabolic dispersion, and appear to be quite heavy. In contrast, the $c$-axis effective mass, $m_z$, required a more elaborate approach due to the FE potential. We determined $m_z$ by solving the one-dimensional Schrödinger equation for a particle of mass $m_z$ in a periodic sawtooth potential (given by $\mathbf{E}_{FE}$) for a bulk crystal composed of 12-layer twins and comparing its spectrum to the DFT-computed $k_z$-dispersions near the conduction and valence band edges (see the "Methods" section and Supplementary Figs. 1, 4). The obtained values for all five materials analysed in this work are listed in Table 1 for electrons, Table 2 and Supplementary Table II for holes.

To analyse self-consistent screening of the underlying FE electric field by the accumulated charge carriers, we assume that all carriers occupy the lowest subband in the resulting quantum well (this assumption is then checked by comparing the computed subband and Fermi energies). This involves solving the Thomas–Fermi problem,

$$\left[-\frac{\hbar^2}{2m_z^{e/h}}\partial_z^2 \mp e\varphi_{e/h}(z)\right]\psi_{e/h}(z) = \epsilon_{e/h}\,\psi_{e/h}(z),$$

$$\partial_z^2\varphi_{e/h}(z) = \pm\frac{en}{\varepsilon_{zz}\varepsilon_0}|\psi_{e/h}(z)|^2,$$

where $\varphi_{e/h}(z)$ is the overall electrostatic potential, and $\psi(z)$ is the wavefunction of the lowest energy bound state in the quantum well at an energy $\epsilon_{e/h}$ for electrons (e–top signs) and holes (h–bottom signs). The electrostatic boundary conditions at mTB, surface, and tTB–both for n- and p-type accumulation layers–are set as $\varphi_{e/h}(0) = 0$, $\partial_z\varphi_{e/h}(0) = \pm\,\mathbf{E}_{FE}$, and $\partial_z\varphi_{e/h}(\infty) = 0$ at the surface ($z = 0$) and in the middle of MX and XM twins ($z \to \infty$), respectively. In contrast, boundary conditions for electron/hole wave functions depend on the type of interface/surface. For a mTB, DFT results suggest that wavefunctions of both electrons and holes are continuous and smooth across the twin boundary, because of matching band edges in MX and XM crystals, hence $\partial_z\psi(0) = 0$ for the lowest-energy state. At the external surface, $\psi(0) = 0$ for both types of charge carriers. Finally, the interlayer twist at a tTB produces different boundary conditions for the two types of charge carriers: for holes with the band edge at the $\Gamma$-point, twist does

## Table 1 | Electron (e) binding energies at mTBs, tTBs and surfaces

| e | $\|E_{FE}\|$ [V/nm] | $n$ [cm⁻²] | $m_z$ [$m_0$] | $\ell$ [nm] | mTB $\epsilon_O$ [meV] | $U_\infty$ [meV] | $\epsilon_b$ [meV] | $\epsilon_F$ [meV] | Surface and tTB $\epsilon_O$ [meV] | $U_\infty$ [meV] | $\epsilon_b$ [meV] | $\epsilon_F$ [meV] |
|---|---|---|---|---|---|---|---|---|---|---|---|---|
| MoS₂ | 0.115 | $0.8 \times 10^{13}$ | 0.51 | 1.09 | 72.4 | 106.4 | 34.0 | 10.0 | 198.8 | 227.8 | 29.0 | 5.0 |
| WS₂ | 0.110 | $0.8 \times 10^{13}$ | 0.48 | 1.13 | 71.7 | 105.4 | 33.7 | 11.1 | 196.9 | 225.7 | 28.8 | 5.5 |
| MoSe₂ | 0.092 | $0.7 \times 10^{13}$ | 0.50 | 1.18 | 62.8 | 92.3 | 29.5 | 10.2 | 172.4 | 197.6 | 25.2 | 5.1 |
| WSe₂ | 0.091 | $0.7 \times 10^{13}$ | 0.43 | 1.25 | 65.6 | 96.4 | 30.8 | 11.9 | 180.0 | 206.3 | 26.3 | 6.0 |
| MoTe₂ | 0.079 | $0.9 \times 10^{13}$ | 0.55 | 1.21 | 55.0 | 80.8 | 25.8 | 9.3 | 150.9 | 173.0 | 22.1 | 4.6 |

Intrinsic FE field $\mathbf{E}_{FE}$, maximum 2D carrier density $n$, $c$-axis effective mass $m_z$, characteristic length of confined states $\ell$, ground state energy $\epsilon_O$, quantum well depth $U_\infty$, binding energy $\epsilon_b$ and Fermi energy $\epsilon_F$ for electrons at the mTBs, tTBs and surfaces.

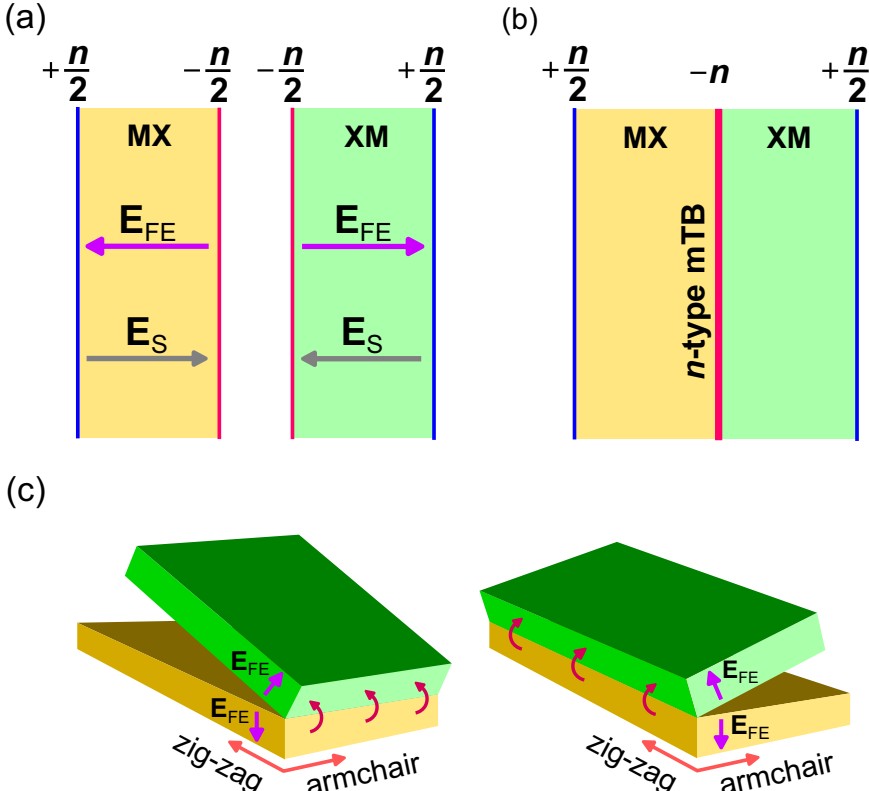

**Fig. 2 | Charge separation in ferroelectric twins.** Sketches illustrate **a** accumulation of opposite charges at the surfaces of MX and XM films, producing a field $\mathbf{E}_S$ that counters the intrinsic FE field $\mathbf{E}_{FE}$. **b** Accumulated charges at external surfaces and twin boundaries in a film with a single n-type mTB. **c** Assembly of n-type TB with parallel (left) and anti-parallel (right) crystallographic axes at the interface by folding a mono-twin against the armchair and the zig–zag direction, respectively.

not obstruct inter-layer hybridisation, so that boundary conditions are the same as for a non-twisted interface, whereas for electrons twist generates a momentum mismatch between Q-point band edges in MX and XM twins which suppresses interlayer hybridisation, separating the quantum well into two—one on the MX and the other on XM side—with $\psi(0) = 0$ at the interface.

To solve this self-consistent problem, we introduce a length, $\ell_{e/h} = \left(\frac{\hbar^2}{e m_z^{e/h} |\mathbf{E}_{FE}|}\right)^{\frac{1}{3}}$, and scaling $z = \ell_{e/h}\xi, \psi_{e/h}(\ell_{e/h}\xi) = \frac{1}{\sqrt{\ell_{e/h}}} f(\xi)$, and $\epsilon_{e/h} = \frac{\hbar^2}{m_z^{e/h}\ell_{e/h}^2} u$, where variables $\xi$ and $u$ and function $f(\xi)$ are dimensionless. Then, we solve numerically the following non-linear integro-differential equation:

$$\left[-\frac{1}{2}\partial_\xi^2 + U(\xi)\right]f(\xi) = u f(\xi), \quad \xi \geq 0,$$
$$U(\xi) = \xi - 2\int_0^\xi d\xi' (\xi - \xi')|f(\xi')|^2, \quad (2)$$
$$f(\infty) = 0, \quad \int_0^\infty d\xi |f(\xi)|^2 = \frac{1}{2}.$$

## Table 2 | Hole (h) binding energies at surfaces

| h | $m_z$ [$m_0$] | $\ell$ [nm] | $\epsilon_0$ [meV] | $U_\infty$ [meV] | $\epsilon_b$ [meV] | $\epsilon_F$ [meV] |
|---|---|---|---|---|---|---|
| MoS$_2$ | 0.88 | 0.91 | 164.8 | 188.9 | 24.1 | 12.6 |
| WS$_2$ | 0.88 | 0.91 | 164.8 | 188.9 | 24.1 | 12.6 |
| MoSe$_2$ | 1.30 | 0.86 | 125.4 | 143.7 | 18.3 | 10.8 |
| WSe$_2$ | 1.10 | 0.91 | 131.6 | 150.9 | 19.2 | 11.7 |
| MoTe$_2$ | 2.09 | 0.77 | 96.7 | 110.8 | 14.1 | 9.8 |

Effective masses along c-axis, characteristic length of confined states, ground state energy, quantum well depth, binding energy and Fermi energy for holes at surfaces.

The normalisation of $f$ in this equation reflects the sum rule of surface charges illustrated in Supplementary Fig. 2 (note that Eq. (2) is formulated in half-space), and the dimensionless function $U(\xi)$ gives the self-consistent potential profile, $U_{e/h} = \hbar^2 U(\xi)/m_z^{e/h}\ell_{e/h}^2$. This enables us to describe both electrons and holes simultaneously in all FE MX$_2$ materials.

Two solutions of Eq. (2), obtained using the shooting procedure described in Methods, one with boundary condition (I) $\partial_\xi f(0) = 0$ and the other (II) with $f(0) = 0$, are shown in Fig. 3a. These solutions encompass all accumulation layers we discuss in this paper: mTB, surfaces and tTB. Solution I (applicable to electrons at mTB and holes at mTB and tTB), which is shown in the left panel of Fig. 3a, produces $u_0 = 0.577$ (which determines the energy of the ground state in the well), with binding energy determined by a parameter $u_b = 0.271$ (obtained as a difference between $u_0$ and $U_\infty \equiv U(\xi \to \infty)$). Solution II (applicable to electrons at tTB and electrons and holes at surfaces), is shown in the r.h.s. panel of Fig. 3a, and produces $u_0 = 1.584$ and $u_b = 0.232$.

Using scaling rules and material parameters in Supplementary Table I, we may now estimate the binding energy and stability of accumulation layers of electrons and holes at mTB, tTB and surfaces of the 3R-TMDs studied here. To discuss the stability of the 2D accumulation layers, we compare the computed binding energies, $\epsilon_b = U_\infty - \epsilon_0$, with the Fermi energies (Table 1) of 2D carriers with corresponding in-plane masses, $m_x, m_y$ (see Supplementary Table I). Results are gathered in Tables 1 and 2 for the more stable electron and surface hole layers, respectively (see Supplementary Table II for holes at mTB and tTB). Here we note that a typical extent of the computed localised states is about $3\ell \sim 3$ nm, which covers about six layers: this justifies the use of a continuum description in the confinement problem. In these estimates, we account for the $g_h = 2$ spin-degeneracy factors for $\Gamma$-point holes and $g_e = 6$ spin-polarised Q valleys for electrons at mTB and surface states. We also note that $g_e = 12$ for electrons at tTBs, reflecting

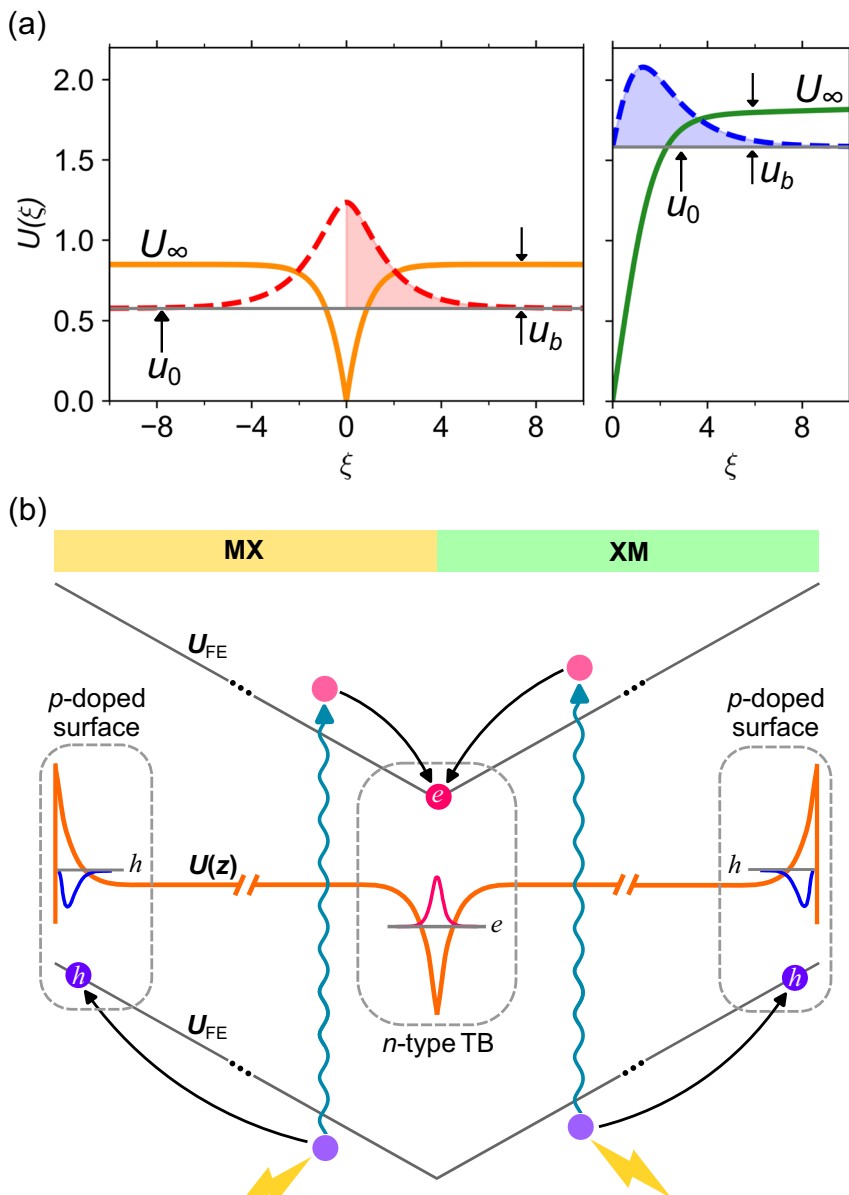

**Fig. 3 | Thomas–Fermi solutions and charge separation. a** Potential profiles resulting from the self-consistent solution of the Thomas–Fermi problem with boundary conditions $\partial_\xi f(0) = 0$ (left panel) and $f(0) = 0$ (right panel). Dimensionless parameters for the ground state ($u_0$), potential well depth ($U_\infty$), and binding energy ($u_b$) are marked on each profile. Dashed lines are solutions for $f$, with a spread $\xi \sim 3$ in both cases I and II; painted areas indicate the $\frac{1}{2}$ normalisation. **b** Separation of photo-excited electrons and holes in a twinned 3R-MX$_2$ film, producing n-type doping mTB (tTB) and p-doped surfaces and a screened (by charge separation) potential profile, $U(z)$.

the contribution from two sets of mismatched Q points in the misaligned MX- and XM-stacking crystals. Also, we note that in all cases, the next level in the quantum well is pushed up into the continuum spectrum, which justifies the use of the assumption that all electrons/holes are in the lowest subband of the resulting quantum well.

When we compare the computed values of $\epsilon_b$ and $\epsilon_F$ in Table 1, we find that the higher degeneracy factor for electrons makes their accumulation at both mTB and surfaces stable at cryogenic temperatures, $T < 100$ K. In contrast, for holes at mTB (see Supplementary Table II), $\epsilon_b \approx \epsilon_F$, making their evaporation into bulk and potentially resulting in a temperature-dependent multi-subband accumulation layer (essentially, a 3D system). However, for the p-type surface states (see Table 2), $\epsilon_b - \epsilon_F \sim 10$ meV for all materials, suggesting that a thin film with a single MX/XM twin boundary would be an ideal host for a 2D electron channel. This system is illustrated in Fig. 3b, where we also suggest the use of photo-excitation of free electron–hole pairs[15,16] to

induce the mTB-surface charge separation, thus, avoiding a dependency of the 2D channel formation on the material doping.

An even more exciting opportunity is offered by an n-type twisted twin boundary (tTB) in a film assembled from two oppositely polarised mono-twin films with misaligned orientation of crystalline axes. As compared to bulk vdW crystals, where twinned structures are formed by chance during the non-equilibrium conditions of crystal growth, the polarisation of domains and orientation of the MX/XM boundary in twistronic structures can be engineered in the course of their assembly. In this case, the control of the twist angle (e.g., by 'folding' a mono-twin film against a properly chosen axis, see Fig. 2c) gives access to moiré patterns with potential for commensurability between their periodicity and carrier density, $n$. Another advantage of the assembled structures is that graphene, embedded at the ends, can be used to contact the electrons at the tTB, and graphene top/bottom gates can be added to access fine-tuning of the tTB carrier density.

For twistronic films assembled with an almost parallel orientation (P) of unit cells, e.g., as if folding a mono-twin crystal against its armchair direction, the superlattice at the twisted interface would be analogous to a triangular lattice of alternating local XM and MX stacking areas, essentially, relocating the twin boundary up and down by one layer. Then, a natural commensurability condition for such a superlattice corresponds to two electrons per moiré supercell which is realised at a 'magic' twist angle $\theta_P^*$, with values for the studied materials listed in Table 3. For a film assembled by 'folding' a mono-twin film against its zig-zag axis, producing an almost antiparallel orientation (AP) of unit cells—rotation angle' $60° + \theta_{AP}$—the superlattice at the tTB would be analogous to a triangular lattice with periodically appearing 2H-stacking areas. In that case, the commensurability condition would correspond to one electron per moiré superlattice unit cell, which would be provided by a larger 'magic' twist angle $\theta_{AP}^*$ (see Table 3 for the values in different materials).

Also, Table 3 contains values of the Wigner parameter, $r_s = \frac{e^2}{4\pi\varepsilon\varepsilon_0\hbar^2}\sqrt{\frac{m_x m_y}{\pi n}}$, which characterises the strength of electron–electron interaction in 2D metals[28]. To estimate it, we used effective masses listed in Supplementary Table I, and effective dielectric constant, $\tilde{\varepsilon} \equiv \sqrt{\varepsilon_\parallel \varepsilon_{zz}}$, for inner interfaces (with the in-plane component[27] of the dielectric tensor $\varepsilon_\parallel$ of TMDs), whereas for surface states we assumed an hBN encapsulation and used $\tilde{\varepsilon}^{-1} = (\tilde{\varepsilon}_{TMD}^{-1} + \tilde{\varepsilon}_{hBN}^{-1})/2$. Due to the large doping densities, we find $2 < r_s < 5.6$, which suggests that interactions are strong but insufficient to crystallise electrons on their own. However, for tTB electrons, one can use the magic twist angle, $\theta_{P/AP}^*$, to promote electrons' Wigner crystallisation with (natural for 2D systems[29–32]) triangular lattice by making electron density commensurate with the moiré superlattice at the corresponding P/AP interface.

One can further extend the analysis of twistronic twin boundaries by considering heterostructures assembled of mono-twin films of different 3R-TMDs. Taking n-type tTB as an example, now, the band edge mismatch, $\gamma_c$, between different semiconductors would lead to electron transfer from the TMD-1 with the higher conduction band edge to TMD-2, with the lower band edge. Similarly to what is illustrated in Fig. 2a, b, the accumulated charge density at the interface would be given by $\tilde{n} = \frac{1}{2}(n_1 + n_2)$, where $n_1$ and $n_2$ are listed for individual materials in Table 1. Then, the analysis of the density profile and quantum well parameters can be carried out using Eq. (2) with boundary condition $f(0) = 0$ and recalculated scaling length,

$$\ell_e = \left[ \frac{\hbar^2}{em_z^{(2)}\left(|\mathbf{E}_{FE}^{(2)}| + \frac{\varepsilon_{zz}^{(1)}}{\varepsilon_{zz}^{(2)}}|\mathbf{E}_{FE}^{(1)}|\right)} \right]^{\frac{1}{3}}.$$

Using the values of dimensionless $u_0 = 1.584$ and $u_b = 0.232$ and TMD parameters in Table 1 and Supplementary Table I, and recent literature[33–35], we identify WSe$_2$/MoS$_2$ and WSe$_2$/MoTe$_2$ as two representative examples of such heterostructures, where the Fermi level of electrons accumulated in MoS$_2$ and MoTe$_2$ would be below the band edge in WSe$_2$. In particular, for WSe$_2$/MoS$_2$, we estimate $\tilde{n} = 0.75 \times 10^{13}$ cm$^{-2}$, $\epsilon_b - \epsilon_F = 35.4$ meV, and $\theta_{P/AP}^* = 4.1°$; for WSe$_2$/MoTe$_2$, $\tilde{n} = 0.8 \times 10^{13}$ cm$^{-2}$, $\epsilon_b - \epsilon_F = 24.1$ meV, and $\theta_{P/AP}^* = 2.4°$. For heterostructures of other pairs of semiconducting 3R-TMDs, the available literature does not agree on exact band edge mismatch values, with the reported data[33–35] indicating that the charge transfer at the surface would be incomplete, so the total accumulated density $\tilde{n} = \frac{1}{2}(n_1 + n_2)$, would be somehow shared between two accumulation layers on the two sides of the tTB. To account for this uncertainty, in the "Methods" section, we describe a recipe for determining how these densities would be shared between two accumulation layers, and offer one characteristic example.

**Table 3 | Wigner parameter and magic twist angles estimated for 3R-MX$_2$ (M = Mo, W; X = S, Se, Te)**

| e | $r_s$ | | $\theta_P^*/\theta_{AP}^*$ |
|---|---|---|---|
| | mTB & tTB | Surface | |
| MoS$_2$ | 2.4 | 5.4 | 3. 3°/4. 7° |
| WS$_2$ | 2.4 | 5.2 | 3. 2°/4. 5° |
| MoSe$_2$ | 2.0 | 4.9 | 3. 4°/4. 8° |
| WSe$_2$ | 1.8 | 4.2 | 3. 3°/4. 7° |
| MoTe$_2$ | 1.8 | 5.6 | 4. 1°/5. 8° |

Twist angles, $\theta_{P/AP}^*$, that would provide commensurability between moiré superlattice and charge carrier density for both types (P and AP) tTBs and Wigner parameter for electrons bound at various twin boundaries and surfaces discussed in this paper.

Overall, we propose a new type of 2D electron system, formed by the accumulation of heavy electrons at twin boundaries in ferroelectric van der Waals semiconductors, such as rhombohedral transition metal dichalcogenides, MX$_2$. The presented analysis also highlights the prospects offered by twistronic structure, assembled by combining two mono-twin films, both of the same materials or heterostructures, for creating strongly correlated states of 2D electrons at the magic-angle twisted twin boundaries.

## Methods

### Structural relaxation

DFT calculations were performed using both bulk 2H and 3R TMDs. For 3R-MX$_2$, mTB structures were created by iterated shifting of adjacent parallel layers in bulk by $\pm a_0/\sqrt{3}$, with sign reversed between the two domains, which was performed for domains with 6, 9 and 12 layers in each domain. Optimisation was performed until energies and forces had converged to a tolerance of $10^{-4}$ and $10^{-3}$ Ry/$a_0$, respectively. A dense $k$-point grid of $22 \times 22 \times 4$ was used to sample 2H bulk structures, while a $22 \times 22 \times 1$ grid was used for mTB structures with large $c$-axis DFT supercell dimensions. As shown in Supplementary Table I, the $c$-axis (interlayer) lattice constants of MX$_2$ twin boundaries are comparable to the 2H-MX$_2$ interlayer lattice constant.

### Intrinsic electric field and out-of-plane masses of electrons and holes

We calculate the ferroelectric field inside twins by approximating the staircase of interlayer potential steps for 6-, 9- and 12-layer domains (see Fig. 1) as $U = e|\mathbf{E}_{FE}|z$. This gives us $|\mathbf{E}_{FE}| \approx 11.5/11.0/9.2/9.1/7.9$ mV/Å for MoS$_2$/WS$_2$/MoSe$_2$/WSe$_2$/MoTe$_2$, respectively, in agreement with the earlier calculated double layer potentials in bilayers[10].

After this, electronic band structures for these multilayers (computed taking into account spin–orbit coupling) were used to determine in-plane ($m_x$, $m_y$—by parabolic fitting) and out-of-plane ($m_z$) effective masses. The $c$-axis mass, $m_z$, was determined using the energies of several lowest minibands in periodic mTBs with 6, 9 and 12 layers in each twin (examples for a $6 + 6$ superstructure are shown in Supplementary Fig. 4 for VB and CB, respectively). Non-dispersive $kHz$-bands were compared between DFT and the energy spectrum of a triangular potential well with a slope corresponding to the intrinsic field $E_{FE}$, which was used to extract the expected effective mass $m_z$ in the equivalent sawtooth potential. This also reproduced $k_z$-dispersion for higher energy bands (see Supplementary Fig. 4). The same procedure was implemented for all studied TMDs. The obtained c-axis masses, $m_z$, are ~10% lighter than the corresponding $c$-axis masses in bulk 2H crystals.

### Multi-parameter shooting method to solve Thomas–Fermi problem

Solutions for the integro-differential equation (2) in the main text are found by employing a shooting method: that is, we integrate it from

relevant boundary conditions ($\partial_\xi f(0) = 0$ and $f(0) = f_0$ for case I, and $f(0) = 0$ and $\partial_\xi f(0) = f'_0$ for case II) at 0 up to $\xi_{max} = 15$, varying the parameter $u$ until we reach a solution that acquires $f(\xi_{max}) = 0$ at the end of the interval without changing sign within it.

Due to the non-linearity of this equation, the initial values of $f_0$ and $f'_0$ are not irrelevant free parameters as it would be for a linear Schrödinger equation. Here, they are involved in determining the normalisation factor of the function $f$, $\mathcal{N} = \int_0^\infty d\xi\,|f|^2$. Therefore we continuously vary the parameters $f_0$ and $f'_0$ with small steps and identify their values for which the normalisation factor becomes $\mathcal{N} = \frac{1}{2}$ (see Supplementary Fig. 2).

Finally, we verify the shooting results by using a finite difference method, where the problem is solved iteratively: the solution $f_i(\xi)$ to the triangular potential $U(\xi) = \xi$ is taken as the starting point to compute a new potential $U(\xi)$, from which a self-consistent cycle follows until convergence is reached in the value of $u$. The evolution of $u$ as a function of the number of iterations for case I is shown as an inset in Supplementary Fig. 5a, where the convergence value $u = 0.577$ agrees exactly with the one obtained via the shooting method.

### Charge redistribution at a heterostructure twin boundary

For twin boundaries assembled from different 3R-TMDs, the band edge mismatch $\gamma_e$ leads to charge transfer from TMD-1 to TMD-2. This results in the redistribution of the total density $\tilde{n} = \frac{1}{2}(n_1 + n_2)$ at the TB as $\tilde{n}_1 = \frac{1}{2}(1 - \nu)n_1$ on TMD-1 and $\tilde{n}_2 = \frac{1}{2}(\nu n_1 + n_2)$ on TMD-2, where $0 \leq \nu \leq 1$. We note that, for the heterostructures described in the main text, the band offset dominates over the binding energies at the mono-twin surfaces, such that we have $\nu = 1$ for both cases. However, when these two energies are comparable, the value of $\nu$ has to be determined from the balance of the Fermi energy on both sides of the TB,

$$\gamma_e + \frac{\hbar^2 u_0}{m_z^{(1)}(\ell_e^{(1)})^2} + \frac{\pi\hbar^2(1-\nu)n_1}{6\sqrt{m_x^{(1)}m_y^{(1)}}} = \frac{\hbar^2 u_0}{m_z^{(2)}(\ell_e^{(2)})^2} + \frac{\pi\hbar^2(\nu n_1 + n_2)}{6\sqrt{m_x^{(2)}m_y^{(2)}}},$$

with

$$\ell_e^{(1)} = \left[\frac{\hbar^2}{e m_z^{(1)}(1-\nu)|\mathbf{E}_{FE}^{(1)}|}\right]^{\frac{1}{3}},$$

$$\ell_e^{(2)} = \left[\frac{\hbar^2}{e m_z^{(2)}\left(|\mathbf{E}_{FE}^{(2)}| + \frac{\varepsilon_{zz}^{(1)}}{\varepsilon_{zz}^{(2)}}\nu|\mathbf{E}_{FE}^{(1)}|\right)}\right]^{\frac{1}{3}},$$

and $u_0 = 1.584$. As an example, we take $WSe_2/WS_2$, where the band edge mismatch of $\gamma_e = 0.22$ eV[35] results in $\nu = 0.74$, indicating that only 26% of the boundary charge remains in $WSe_2$, the rest being transferred to $WS_2$. This scenario is depicted in Supplementary Fig. 5b, showing the different potential profiles and charge density distribution, determined by the electron wave functions $\psi_e(z)$. In particular, for $WSe_2$ we estimate $\tilde{n}_1 = 0.09 \times 10^{13}$ cm$^{-2}$ and $\epsilon_b^{(1)} - \epsilon_F^{(1)} = 7.6$ meV; for $WS_2$, $\tilde{n}_2 = 0.62 \times 10^{13}$ cm$^{-2}$ and $\epsilon_b^{(2)} - \epsilon_F^{(2)} = 26.9$ meV.

### Data availability

The data supporting the findings of this study have been included in the main text and Supplementary Information. All other information can be obtained from the corresponding author upon request.

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

## Acknowledgements

The authors thank Z. Sofer, I. Rozhansky, R. Gorbachev and A. Geim for fruitful discussions. I.S. and X.L. acknowledge financial support from the University of Manchester's Dean's Doctoral Scholarship. This work was supported by the EC-FET Core 3 European Graphene Flagship Project, EC-FET Quantum Flagship Project 2D-SIPC, EPSRC Grants EP/S030719/1 and EP/V007033/1, and the Lloyd Register Foundation Nanotechnology Grant. Calculations were performed using the Sulis Tier 2 HPC platform hosted by the Scientific Computing Research Technology Platform at the University of Warwick and funded by EPSRC Grant EP/T022108/1 and the HPC Midlands+ consortium.

## Author contributions

V.I.F. conceived the idea of the project and designed its methodology. X.L. and J.G.M. performed DFT modelling and determined input parameters for the mesoscale model and the Thomas–Fermi problem, which was solved numerically by I.S. All authors were involved in discussions and contributed equally to writing the manuscript.

## Competing interests

The authors declare no competing interests.
