## [Peer Review File · Nature Communications]

Two-dimensional electrons at mirror and twistrionic twin boundaries in van der Waals ferroelectricsREVIEWER COMMENTS

Reviewer #1 (Remarks to the Author):

The authors proposed a new type of 2D electron systems, formed by the accumulation of heavy electrons at twin boundaries in ferroelectric van der Waals semiconductors, such as rhombohedral MoS₂ and WS₂. The presented analysis also highlights the prospects offered by twistrionic structure, assembled by combining two monotwin flms, for creating strongly correlated states of 2D electrons at the magic-angle twisted twin boundaries. The results may deserve interest, and the following concerns are to be addressed:

1. The density $n=0.8 \times 10^{13} / \text{cm}^2$ was estimated using formula (1), which seems to be independent on layer number. I think the dielectric constant as well as such charge density induced by polar discontinuity should vary with layer number?
2. The system is a semiconductor in ExtData Fig. 3, but the bandgap can be further reduced and even turn metallic for more layers with stronger polar discontinuity. So the charge will also turn from bound charge to free charge?
3. Does the band splittings in ExtData Fig. 3 indicate a net magnetization induced somewhere?

Reviewer #2 (Remarks to the Author):

This paper presents a detailed study on semiconducting transition metal dichalcogenides (MX₂), focusing on their 2H and rhombohedral polytypes, characterized by antiparallel and parallel orientations of monolayer lattices, respectively. The authors analyze rhombohedral MX₂'s bulk form, highlighting its ferroelectric properties and out-of-plane electric polarization dictated by stacking.

A key contribution is the prediction that twin boundaries, which separate adjacent polarization domains with reversed built-in electric fields, can host two-dimensional electrons and holes with an areal density of approximately 10^{13} cm^{-2} . Notably, the modeling suggests that n-doped twin boundaries have a more promising binding energy than p-doped ones, while hole accumulation remains stable at the external surfaces of a twinned film.

The paper also proposes that introducing a twist at the twinned interface with a 'magic' angle of $\theta \approx 3^\circ$ could facilitate strongly correlated states of electrons, such as Wigner crystals, due to the commensurability between the moiré pattern and the accumulated carrier density.

Overall, this paper provides valuable insights into the electronic properties of MX₂ materials, particularly regarding twin boundaries and twisted interfaces. However, the lack of experimental validation and the purely theoretical nature of the work may limit its readership and impact.

Reviewer #3 (Remarks to the Author):

Based on the out of plane ferroelectric feature of 3R-MX₂, this work proposed a scheme to realize 2D electron system localized at the twin boundary between adjacent ferroelectric domains with inverse polarization. First-principles calculations combined with continuum mesoscale model predict that the 2D areal density of electrons can reach $10^{13} / \text{cm}^2$. In addition, the authors also studied the twin-boundary 2D electron system with twist angle and

showed that strongly correlated states such as Wigner crystals can occur. In a word, the authors have designed a new scheme to realize 2D electron system and provided a new platform to study 2D electron system and related phenomena. I think this work is both important and interesting, and hence recommend to publish it in Nature Communications.

Only one comment:

The two words “internal” and “twistronic” in the title are not on the same level, because all twistronic twin boundaries are internal.

Two-dimensional electrons at internal and twistrionic twin
boundaries of van der Waals ferroelectrics
Response Letter

James G. McHugh, Xue Li, Isaac Soltero and Vladimir I. Fal'ko

Contents

Response to Referee 1	2
Response to Referee 2	3
Response to Referee 3	4

Response to Referee 1

Referee 1: The density $n = 0.8 \times 10^{13} \text{ cm}^{-2}$ was estimated using formula (1), which seems to be independent on layer number. I think the dielectric constant as well as such charge density induced by polar discontinuity should vary with layer number?

The system is a semiconductor in Extended Data Fig. 3, but the bandgap can be further reduced and even turn metallic for more layers with stronger polar discontinuity. So the charge will also turn from bound charge to free charge?

For a thick film (layer number $\gg 1$), this would be the same independently of layer number, though the surface accumulation layer would deplete into the substrate if the band gap on the substrate would be smaller than the TMD. On the one hand, this would stabilise the accumulation at the internal TB. On the other hand, tunnelling of carriers into those states in the environment excludes atomically thin films from our consideration.

Indeed, for a thick monotwin flake with $N > E_g/\Delta$, where E_g is the TMD band gap, the surfaces of the flake would be self-doping, with a density $\pm n/2$. For example, for 3R-MoS₂, this would be $N > 1600/70 \approx 25$ layers. To clarify this point, we now explicitly discuss this on page 2, with the following additional text: *"To mention, a thick MX (or XM) monotwin film with $N \gg E_g/\Delta \sim 20 - 30$ layers for the TMDs studied here (where E_g is the semiconductor bandgap) would self-dope its surfaces to the same densities as illustrated in Fig. 2a. Similarly, a multidomain crystal with such thick twins would self-dope with a density distribution as in Fig. 2b."* We have also added part of Ext. Fig. 2 to the main text to create a new Fig. 2 to help illustrate this points.

Referee 1: Does the band splittings in Extended Data Fig. 3 indicate a net magnetization induced somewhere?

There is no net magnetization due to this spin-splitting, as there are six Q-points which occur in pairs which have opposite spin-splitting, reflecting the time inversion symmetry of the system. We have added a diagram of the Brillouin-zone to Extended Data Fig. 3 and discuss this in the associated caption, which now contains the additional text: *"... top-down view of TMD Brillouin zone, illustrating 6 degenerate Q-points which occur as Kramers pairs."*, and *"...spin orientation in spin-split bands is identified by colour, with opposite spin-splitting occurring at the $\pm Q_i$ points."*,

Response to Referee 2

Referee 2: ...Overall, this paper provides valuable insights into the electronic properties of MX_2 materials, particularly regarding twin boundaries and twisted interfaces. However, the lack of experimental validation and the purely theoretical nature of the work may limit its readership and impact.

Our result is a prediction of the formation of a two-dimensional electron system rather than an interpretation of a past experiment. The purpose of this publication will be to stimulate new experimental studies. We note that various rhombohedral TMD crystals are being grown and studied by various experimental groups worldwide, not mentioning that there is commercial supply of MoS_2 , WS_2 and MoSe_2 , provided by the Dutch company HQGraphene.

However, we appreciate the concerns of the Referee in view maximally broadening impact of our predictions of a new type of two-dimensional electron system and its specifics in twistrionic structures. Therefore, in order to broaden the readership, in this revised version, we extend these predictions onto all semiconducting TMDs which have been synthesised in 3R bulk form, now, including MoS_2 , WS_2 , MoSe_2 , WSe_2 and MoTe_2 . This new information has been added to Tables I, II and Extended Data Tables I and II. Moreover, we also make analogous predictions for heterostructures of 3R-TMDs and we describe the characteristic examples in the additional paragraph added on page 5 and a generic recipe of the description of such heterostructures in new subsection “Charge redistribution at a heterostructure twin boundary” in Methods.

Response to Referee 3

Referee 3: ...I think this work is both important and interesting, and hence recommend to publish it in Nature Communications. Only one comment: The two words “internal” and “twistronic” in the title are not on the same level, because all twistronic twin boundaries are internal.

We thank the referee for raising this point. Indeed, it should have read ”mirror and twistronic twin boundaries”. The new title reads: ”Two-dimensional electrons at mirror and twistronic twin boundaries in van der Waals ferroelectrics”.

REVIEWERS' COMMENTS

Reviewer #1 (Remarks to the Author):

The authors have addressed all the concerns and the paper might be accepted

Reviewer #2 (Remarks to the Author):

I am satisfied with revision made the authors.

Reviewer #3 (Remarks to the Author):

The manuscript has been improved according to the comments of all reviewers. I recommend to publish it in Nature Communications now.

We thank the Referees for their favorable assessment of our work.

Dr. James McHugh